# Bleb analysis using anterior segment optical coherence tomography after trabeculectomy with amniotic membrane transplantation

**Hwayeong Kim[1], Sangwoo Moon[1,2], Eunah Kim[1], Jinmi Kim[3], Jiwoong Lee[1,2]\***

**1** Department of Ophthalmology, Pusan National University College of Medicine, Busan, Korea,
**2** Biomedical Research Institute, Pusan National University Hospital, Busan, Korea, **3** Department of
Biostatistics, Clinical Trial Center, Biomedical Research Institute, Pusan National University Hospital, Busan,
Korea

\* alertlee@naver.com

## Abstract

pone.0285127

University Faculty of Medicine, EGYPT

**Data Availability Statement:** All relevant data are
within the paper and its Supporting information
files.

### Introduction

Little has been known about the intrableb structures associated with bleb function after trabeculectomy with amniotic membrane transplantation (AMT). The aim of this study is to analyze the characteristics of intrableb structures using anterior segment optical coherence tomography (AS-OCT) after trabeculectomy with AMT.

### Methods

A total of 68 eyes of 68 patients with primary open-angle glaucoma who underwent trabeculectomy with AMT were included. Surgical success was defined as intraocular pressure (IOP) ≤ 18 mmHg and IOP reduction of ≥ 20% without medication on AS-OCT examination. Intrableb parameters, including bleb height, bleb wall thickness, striping layer thickness, bleb wall reflectivity, fluid-filled space score, fluid-filled space height, and microcyst formation were evaluated using AS-OCT. Logistic regression analysis was performed to determine factors associated with IOP control.

### Results

Of the 68 eyes, 56 eyes were assigned to the success group and 12 eyes to the failure group. In the success group, bleb height ($P = 0.009$), bleb wall thickness ($P = 0.001$), striping layer thickness ($P = 0.001$), fluid-filled space score ($P = 0.001$), and frequency of microcyst formation ($P = 0.001$) were greater than those in the failure group. Bleb wall reflectivity was higher in the failure group than in the success group ($P < 0.001$). In the univariate logistic regression analysis, previous cataract surgery was significantly associated with surgical failure (odds ratio = 5.769, $P = 0.032$).

**Funding:** This research was supported by the Patient Centered Clinical Research Coordinating Center, funded by the Ministry of Health & Welfare, Republic of Korea (https://www.neca.re.kr/) under grant no. HI19C0481, HC19C0276 to JL, SM, and HK. The funders had no role in study design, data collection and analysis, decision to publish, or preparation of the manuscript. There was no additional external funding received for this study".

**Competing interests:** The authors have declared that no competing interests exist.

## Conclusion

A posteriorly extending fluid-filled space, tall bleb with low reflectivity, and thick striping layer were characteristics of successful filtering blebs after trabeculectomy with AMT.

## Introduction

Trabeculectomy is the gold standard for glaucoma filtering surgery [1, 2]. However, postoperative wound healing, which occurs at the episclera, may block aqueous humor outflow, leading to bleb failure [3, 4]. Although antifibrotic agents, including mitomycin-C (MMC) or 5-fluoro-uracil (5-FU), are applied at the episclera to inhibit fibrosis at the filtering site [3, 4], their application is associated with bleb-related complications, such as avascular bleb formation, bleb leak, or corneal erosion [5–8]. An experimental study reported that suberoylanilide hydroxamic acid, a histone deacetylase inhibitor prevented excessive wound healing and scar formation in a rabbit model of glaucoma filtration surgery [9].

Trabeculectomy with amniotic membrane transplantation (AMT) above or under the scleral flap had a high success rate and fewer complications [10–12]. We previously evaluated the effect of AMT on trabeculectomy and found that it was a safe and effective procedure for intraocular pressure (IOP) reduction in patients with primary open-angle glaucoma (POAG) and pseudoexfoliation glaucoma [13, 14]. Furthermore, the incidence of avascular blebs was significantly lower in the AMT group than in the control group. The transplanted amniotic membrane may act as part of the bleb wall and relieve pressure on it [14].

The internal structure of bleb associated with IOP control after trabeculectomy cannot be evaluated using a silt-lamp. Several studies have used ultrasound biomicroscopy (UBM) to investigate intrableb structures [15–17]. Avitabile et al. [15] and Yamamoto et al. [16] reported that better route visibility under the scleral flap and low reflectivity inside the bleb was associated with good IOP control. Jinza et al. [17] reported that the thickness of the aqueous drainage route under the scleral flap correlated with the development of a filtering bleb. Nakamura et al. [18] evaluated intrableb structures after trabeculectomy with AMT using UBM and reported that the extent of the subconjunctival fluid-filled space was associated with long-term IOP control. However, UBM examination requires considerable effort to obtain high-quality images and contact with the eye, which can cause an infection.

Recently, anterior segment optical coherence tomography (AS-OCT) has played a major role in anterior segment imaging. AS-OCT can capture images in a sitting position, unlike UBM, which requires the patients to be in the supine position. In addition, it is a non-contact scanning method that provides a higher axial resolution than UBM [19]. Several studies have evaluated the intrableb structure after glaucoma surgery using AS-OCT [20–24]. Narita et al. [20, 21] reported that taller blebs with thicker hypo-reflective walls were characteristic of a well-functioning post-trabeculectomy bleb. Hamanaka et al. [23] investigated the correlation between bleb morphology and IOP control after trabeculectomy with two different types of conjunctival incisions using AS-OCT. They reported that IOP control after limbal-based trabeculectomy was more dependent on larger and thinner wall blebs than that after fornix-based trabeculectomy. However, these studies analyzed intrableb structure after trabeculectomy without AMT.

To the best of our knowledge, this is the first study to evaluate intrableb structure after trabeculectomy with AMT using AS-OCT. In this study, we investigated the features of the

filtering bleb after trabeculectomy with AMT using AS-OCT and evaluated the factors associated with IOP control in patients with POAG.

## Materials and methods

### Study design and population

The medical records of patients with POAG who underwent fornix-based trabeculectomy with AMT between November 2014 and March 2020 at the Pusan National University Hospital were retrospectively reviewed. The study was approved by the institutional review board of Pusan National University (approval no. 2211-012-121) and was conducted in accordance with the tenets of the Declaration of Helsinki. All patients provided written informed consent for the surgical procedures, storage of their information in the hospital database, and research.

The inclusion criteria were as follows: (1) IOP not controlled appropriately despite maximally tolerated medical therapy, laser trabeculoplasty, or both; and (2) those allergic to or with side effects from glaucoma medication. The following patients were excluded: (1) those with secondary glaucoma and other ocular or systemic diseases affecting the optic nerve head, macula, or visual field; and (2) patients who underwent ocular surgery for any reason other than uncomplicated cataract extraction. If trabeculectomy with AMT was performed bilaterally, the first eye was enrolled.

All the patients underwent thorough ophthalmologic examination, including best corrected visual acuity, slit-lamp examination, IOP measurement using Goldmann applanation tonometry, gonioscopy, fundus examination, stereoscopic optic disc and red-free retinal nerve fiber layer photography (AFC-210; Nidek, Aichi, Japan), biometry using the IOL Master (Carl Zeiss Meditec, Dublin, CA, USA), and standard automated perimetry using a Humphrey Visual Field Analyzer 750i instrument (Carl Zeiss Meditec, Dublin, CA, USA) and the Swedish interactive threshold algorithm (SITA) standard C24-2. Central corneal thickness (CCT) was measured using ultrasonic pachymetry (Pachmate; DGH Technology, Exton, PA, USA). Keratometry was performed using an auto-kerato-refractometer (ARK-510A; NIDEK, Hiroshi, Japan). All patients underwent AS-OCT examination at least six months after trabeculectomy. Surgical success was defined as an IOP $\leq$ 18 mmHg and an IOP reduction $\geq$ 20% without glaucoma medication at the time of AS-OCT examination [25].

### Surgical procedure

All surgeries were performed by an experienced glaucoma surgeon (J. L.). Under local anesthesia, a 5–6 mm fornix-based conjunctival peritomy was performed. The Tenon's capsule was separated from the sclera. A trapezoidal scleral flap (basal 4.5 mm, apical 2.75 mm, bilateral 2.75 mm) with 2/3 of the sclera thickness was created. Weck-cell sponges soaked in 0.4 mg/mL (0.04%) of MMC were placed between the Tenon's capsule and sclera for 2–3 minutes. Subsequently, the sponges were removed and the space was washed with 20 mL of balanced salt solution. An inner sclerostomy was performed underneath the scleral flap, followed by peripheral iridectomy. The scleral flap was sutured with two preplaced releasable sutures using 9–0 nylon (Ethicon Inc., Johnson & Johnson, Somerville, NJ, USA). A single layer of cryopreserved amniotic membrane (MS Amnion, MS BIO Inc., Seongnam, Korea) of size 15 × 15 mm was placed beneath the Tenon's capsule with the stromal side facing up. The amniotic membrane was fixed to the lateral side of the scleral flap using two interrupted 10–0 nylon sutures (Ethicon Inc., Johnson & Johnson). The conjunctiva was sutured to the sclera together with Tenon's capsule using 10–0 nylon sutures. The anterior chamber was injected with a balanced salt solution through the corneal puncture site, and the degree of aqueous outflow through the scleral flap and bleb leakage through the conjunctival sutures were assessed (Fig 1). After

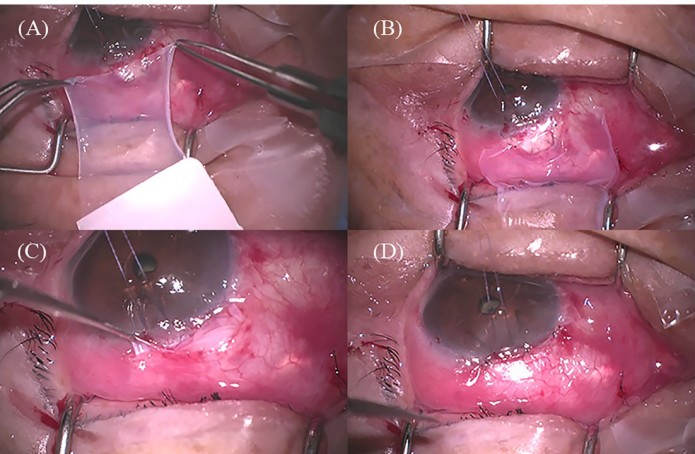

**Fig 1. Surgical procedure of trabeculectomy with amniotic membrane transplantation.** (A) Amniotic membrane was peeled from the nitrocellulose membrane. (B) Amniotic membrane was placed over the scleral flap with the stromal side facing up. The limbal side of the amniotic membrane was secured to both sides of the scleral flap margin with two 10–0 nylon sutures. (C) Amniotic membrane was placed between the Tenon's capsule and scleral flap. (D) The conjunctiva and Tenon's capsule were closed with interrupted 10–0 nylon sutures.

trabeculectomy, levofloxacin (Cravit®, Santen Pharm, Co., Osaka, Japan) was instilled four times a day and prednisolone acetate (Predbell®, CKD Pharm, Co., Seoul, Korea) was instilled six times a day for a month and then tapered every 8–12 weeks according to bleb morphology and IOP.

## Anterior segment optical coherence tomography imaging

Post-operative bleb images were obtained using the anterior segment module (volume scan vertical-filtering bleb mode, "VolBleb") of the Spectralis OCT (Heidelberg Engineering, Heidelberg, Germany), which uses a shorter wavelength light source (870 nm) and higher speed (40,000 A-scans/second) than time domain-OCT. The VolBleb mode was used with enhanced depth imaging and automatic real-time images at 16 frames. The OCT scan pattern size was 8.3 × 2.8 mm, and the number of B-scans was 21 with a 139 μm-distance between the scans. The penetration depth was 1.9 mm with a lateral resolution scaling of 10.84 μm/pixel and an axial resolution scaling of 3.87 μm/pixel. Images with a quality score > 25 dB were included in the analysis.

Intrableb structural parameters were measured using the device's built-in software Heidelberg Eye (Version 1.10.2.0; Heidelberg Engineering, Heidelberg, Germany), and bleb wall reflectivity was measured with ImageJ (ImageJ 1.50b, http://imagej.nih.gov/ij/; developed by Wayne Rasband, National Institutes of Health, Bethesda, MD, USA) [22].

Horizontal (tangential to the limbus) and vertical (radial perpendicular to the limbus) scans were taken at the maximum elevation of each eye bleb. The quantitative parameters included the maximum bleb height, bleb wall thickness, striping layer thickness, fluid-filled space height, fluid-filled space score (FFSS), and bleb wall reflectivity. The mean horizontal and vertical measurements were used for the analysis.

Bleb height was measured as the maximal vertical distance between the first reflective signal from the conjunctiva and a straight line perpendicular to the tangent to the sclera. Manual adjustment of the contrast setting enabled the identification of the scleral edge when the margin was not apparent in the OCT image. Bleb wall thickness consisting of the conjunctiva,

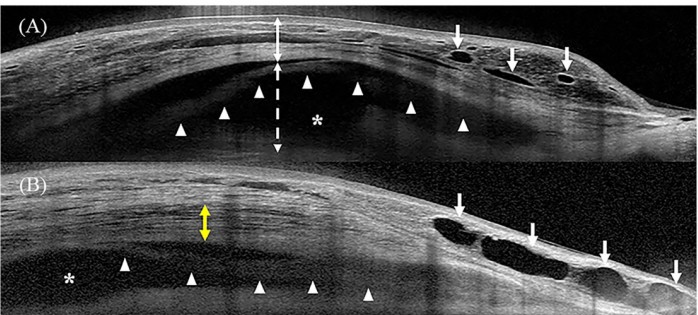

**Fig 2. Intrableb structure measurements.** (A) Vertical anterior segment optical coherence tomography (AS-OCT) scan of the bleb (B) Horizontal AS-OCT scan of the bleb. The *white continuous* and *dotted two-way arrows* indicate the bleb wall thickness and fluid-filled space height, respectively. The *white arrow* indicates the microcyst, and the *asterisk* indicates the fluid-filled space. The *white arrow heads* indicate the visible amniotic membrane beneath the bleb wall. The *yellow two-way arrow* indicates the hyporeflective layers with striping phenomenon.

Tenon's capsule, and/or incorporated amniotic membrane was measured as the maximal vertical distance between the first reflective signal of the conjunctiva and the top of the fluid-filled space. The striping layer was defined as multiple parallel and fluid-filled channels in Tenon's capsule, which resembled a honeycombed structure (Fig 2) [20, 26].

The fluid-filled space height was measured as the maximal vertical distance in the signal void or hyporeflective area between the bottom of the inner bleb wall and the top of the sclera along a straight line perpendicular to the tangent of the sclera (Fig 2). The FFSS was graded from 0 to 2 as follows: (1) score 0, no visible fluid-filled space; (2) score 1, limited and demarcated fluid-filled space with a clear posterior margin; and (3) score 2, diffuse fluid-filled space extending posteriorly beyond the view of the image field (Fig 3) [18, 21].

Bleb wall reflectivity was analyzed using ImageJ software. An ellipse mark of the background near the bleb wall and three evenly spaced ellipse marks in the bleb wall (anterior, middle, and posterior) were placed to measure the bleb wall reflectivity. The background value was subtracted from the three reflectivity values; subsequently, the average of the three bleb wall reflectivity values was calculated [22].

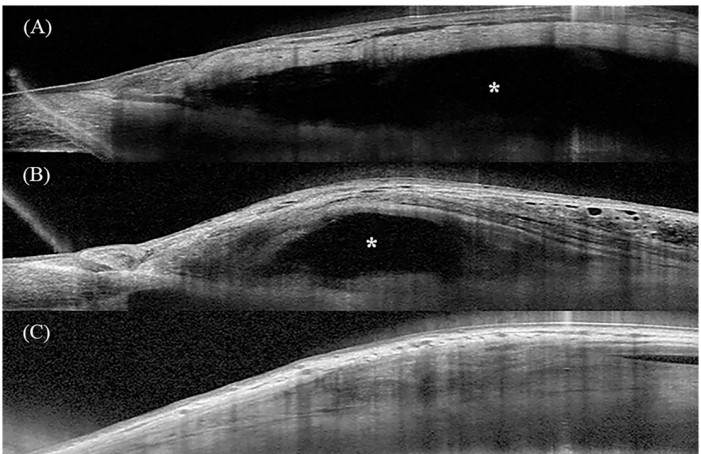

**Fig 3. Representative anterior segment optical coherence tomographic images of the fluid-filled space score (FFSS).** (A) FFSS 2. Fluid-filled space is diffuse and extends posteriorly beyond the field of the image view. (B) FFSS 1. Fluid-filled space is limited. (C) FFSS 0. No fluid-filled space is observed. *Asterisks* indicate fluid-filled spaces.

Microcyst formation was a qualitative parameter that was defined as a hyporeflective or signal void-space within or beneath the epithelial layer of the bleb wall [20].

### Statistical analyses

Data distribution normality was checked using the Kolmogorov-Smirnov test. Differences between the success and failure groups were analyzed using the Mann-Whitney U- test or independent-sample t-test for continuous variables, and the Chi-squared or Fisher's exact test for categorical variables. Univariate logistic regression analysis was used to determine the baseline risk factors associated with surgical failure, and the odds ratios for these parameters were calculated. To examine relationships between variables, we performed Pearson's correlation and point biserial correlation analysis for parametric tests, and Spearman's correlation analysis for non-parametric tests. To evaluate the differences over time, a linear regression model was used to test the interaction effect, and the time effect was adjusted. All statistical analyses were performed using the language R (version 4.0.5; R Project for Statistical Computing, Vienna, Austria). $P$-values $< 0.05$ were considered statistically significant.

## Results

### Study population

A total of 68 eyes of 68 patients (49 males and 19 females) who underwent trabeculectomy with AMT were included. Of the 68 eyes, 56 eyes were included in the success group and 12 eyes were included in the failure group. Mean age at the time of surgery was 61.25 ± 12.44 years and 61.74 ± 14.06 years for the success and failure groups, respectively. The interval between surgery and AS-OCT examination ranged from 6 to 39 months (median, 19 months). The interval was 19.94 ± 9.29 months and 23.59 ± 10.43 months in the success and the failure groups, respectively ($P = 0.231$).

IOP at AS-OCT examination was 11.59 ± 3.20 mmHg and 21.25 ± 4.99 mmHg in the success and the failure groups, respectively ($P < 0.001$). The number of pseudophakic eyes before surgery was lower in the success group (46.4%) than in the failure group (83.3%) ($P = 0.026$). There were no significant differences in sex, eye laterality, preoperative IOP, number of preoperative medications, preoperative visual acuity, CCT, axial length, and visual field parameters between the success and failure groups ($P \geq 0.128$ for all) (Table 1).

### AS-OCT images of the filtering bleb

The intrableb structures evaluated using AS-OCT were compared between the success and failure groups (Table 2 and Fig 4). No significant interaction effect was found between the AS-OCT examination time and groups for all bleb parameters (S1 Fig, $P \geq 0.333$ for all). As the AS-OCT examination was performed at different times for each patient, the effect of examination time was adjusted (Table 2). In the success group, bleb height and bleb wall thickness, striping layer thickness, and FFSS were greater than those in the failure group (unadjusted $Ps \leq 0.009$; adjusted $Ps \leq 0.015$). Bleb wall reflectivity was lower in the success group than that in the failure group (unadjusted $P < 0.001$; adjusted $P < 0.001$). Microcyst formation was more frequently seen in the success group than in the failure group (unadjusted $P = 0.001$; adjusted $P = 0.002$). There was no significant difference in the fluid-filled space height between the two groups (unadjusted $P = 0.797$; adjusted $P = 0.801$) (Fig 4).

**Table 1. Demographics and clinical characteristics of patients.**

| | Overall | Success | Failure | p-value |
|---|---|---|---|---|
| Number of patients | 68 | 56 (82.4) | 12 (17.6) | |
| Sex, Female | 19 (27.9) | 17 (30.4) | 2 (16.7) | 0.487 |
| Diabetes mellitus | 12 (17.6) | 11 (19.6) | 1 (8.3) | 0.677 |
| Hypertension | 22 (32.4) | 16 (28.6) | 6 (50.0) | 0.182 |
| Age at trabeculectomy, year | 61.34 ± 12.63 | 61.25 ± 12.44 | 61.74 ± 14.06 | 0.797 |
| Eye laterality, right | 29 (42.6) | 24 (42.9) | 5 (41.7) | 1.000 |
| Preoperative IOP, mmHg | 32.16 ± 9.81 | 32.09 ± 9.11 | 32.50 ± 13.08 | 0.896 |
| Interval between surgery and AS-OCT, months (range) | 20.58 ± 9.53 (6 to 39) | 19.94 ± 9.29 (6 to 39) | 23.59 ± 10.43 (6 to 39) | 0.231 |
| IOP at AS-OCT test, mmHg | 13.29 ± 5.13 | 11.59 ± 3.20 | 21.25 ± 4.99 | 0.000 |
| No. of preoperative medications | 4.03 ± 0.60 | 3.98 ± 0.62 | 4.25 ± 0.45 | 0.121 |
| Preoperative visual acuity, logMAR | 0.32 ± 0.41 | 0.34 ± 0.46 | 0.28 ± 0.46 | 0.584 |
| Preoperative lens status | | | | 0.026 |
| Phakia | 32 (47.1) | 30 (53.6) | 2 (16.7) | |
| Pseudophakia | 36 (52.9) | 26 (46.4) | 10 (83.3) | |
| Central corneal thickness, μm | 533.50 ± 45.25 | 532.14 ± 47.24 | 539.83 ± 35.51 | 0.597 |
| Axial length, mm | 24.72 ± 1.92 | 24.49 ± 1.50 | 25.80 ± 3.13 | 0.128 |
| Spherical equivalent, diopter | −1.85 ± 2.73 | −1.72 ± 2.56 | −2.47 ± 3.46 | 0.380 |
| Visual field parameter | | | | |
| Visual Field Index, % | 50.96 ± 31.81 | 51.68 ± 32.38 | 47.58 ± 30.10 | 0.689 |
| Mean deviation, dB | −17.38 ± 9.21 | −17.12 ± 9.42 | −18.56 ± 8.44 | 0.626 |
| Pattern standard deviation, dB | 7.60 ± 3.76 | 7.38 ± 3.78 | 8.65 ± 3.63 | 0.289 |

Counting fingers at 30 cm was considered equivalent to Snellen value of 20/2000, which corresponds to a logMAR value of 2.0. Hand motion acuity was considered equivalent to a Snellen value of 20/20,000, which corresponds to a logMAR value of 3.0. Values are presented as mean ± standard deviation or number (%) unless otherwise indicated.

AMT, amniotic membrane transplantation; AS-OCT, anterior segment optical coherence tomography; IOP, intraocular pressure; logMAR, logarithm of the minimum angle of resolution.

## Baseline risk factors

Univariate logistic regression analysis was performed to determine preoperative clinical characteristics associated with surgical failure (Table 3). Previous cataract surgery was significantly associated with surgical failure (odds ratio = 5.769, $P = 0.032$). Age, sex, preoperative IOP, the

**Table 2. Intrableb parameters assessed with anterior segment optical coherence tomography after trabeculectomy with amniotic membrane transplantation.**

| Intrableb parameters | Overall (n = 68) | Success (n = 56) | Failure (n = 12) | p-value | p-value* |
|---|---|---|---|---|---|
| Bleb height, μm | 1396.51 ± 310.98 | 1441.43 ± 272.89 | 1186.92 ± 398.13 | 0.009 | 0.015 |
| Bleb wall thickness, μm | 745.00 ± 263.57 | 794.29 ± 253.55 | 515.00 ± 177.78 | 0.001 | 0.001 |
| Striping layer thickness, μm | 249.85 ± 218.10 | 290.52 ± 217.26 | 60.08 ± 77.51 | 0.001 | 0.001 |
| Bleb wall reflectivity | 106.96 ± 28.06 | 98.99 ± 23.42 | 144.16 ± 14.85 | <0.001 | <0.001 |
| Fluid-filled space score | 1.60 ± 0.54 | 1.70 ± 0.44 | 1.12 ± 0.71 | 0.001 | 0.001 |
| Fluid-filled space height, μm | 651.51 ± 300.08 | 647.14 ± 521.20 | 671.92 ± 482.10 | 0.797 | 0.801 |
| Microcyst formation | 58 (85.3) | 52 (92.9) | 6 (50.0) | 0.001 | 0.002 |

Values are presented as mean ± standard deviation or number (%) unless otherwise indicated.

*Adjusted for AS-OCT examination time.

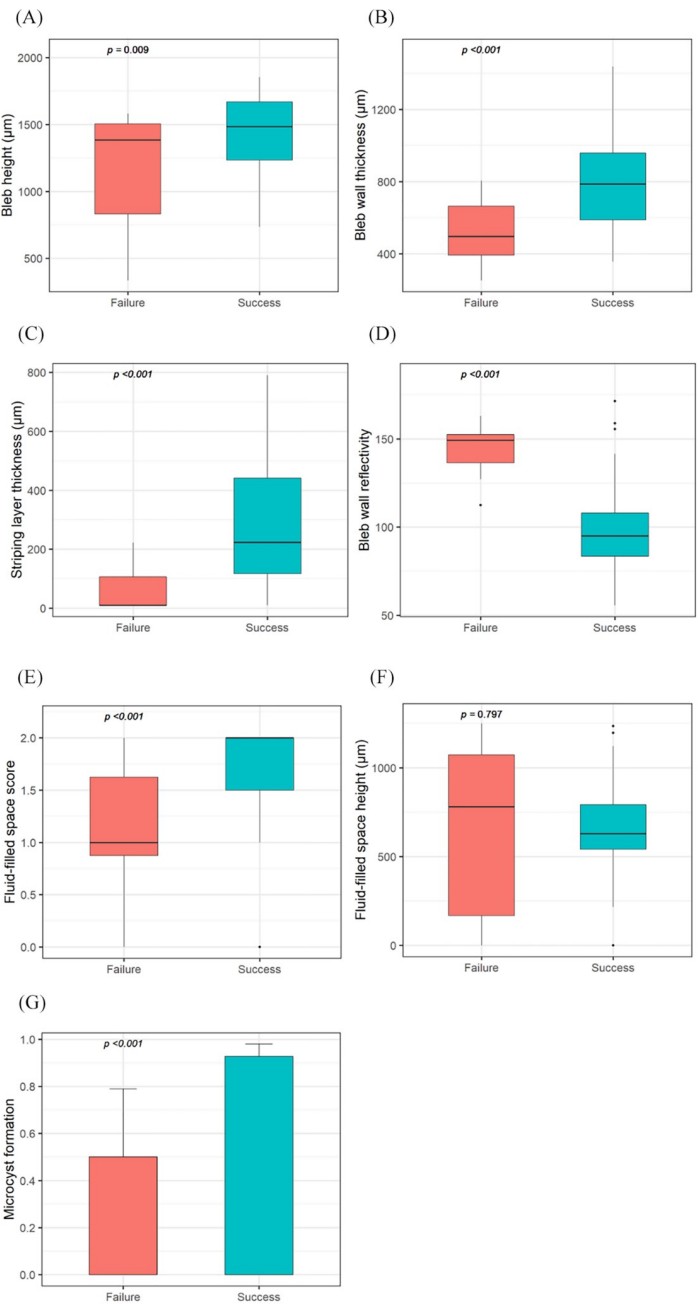

**Fig 4. Boxplot of the intrableb parameters assessed with anterior segment optical coherence tomography after trabeculectomy with amniotic membrane transplantation for the comparison of the success and failure groups.** (A) Bleb was higher in the success group than that in the failure group. (B) Bleb wall was thicker in the success group than that in the failure group. (C) Stripping layer was thicker in the success groups than that in the failure group. (D) Bleb wall reflectivity was lower in the success group than that in the failure group. (E) Fluid-filled space score was greater than that in the failure group. (F) There was no significant difference in the fluid-filled space height between the two groups. (G) Microcyst formation was more frequently noted in the success group than in the failure group.

number of preoperative medications, preoperative visual acuity, CCT, axial length, and visual field parameters were not associated with surgical failure.

The effect of duration between cataract extraction and trabeculectomy with AMT on surgical failure and relationship between the duration and intrableb structures were evaluated. The

**Table 3. Baseline risk factors associated with surgical failure after trabeculectomy with amniotic membrane transplantation.**

| | Univariate logistic regression analysis | |
| --- | --- | --- |
| | Odds ratio (95% CI) | *p*-value |
| Age, year | 1.003 (0.96, 1.064) | 0.902 |
| Sex, female vs. male | 0.459 (0.066, 1.984) | 0.346 |
| Preoperative IOP, mmHg | 1.004 (0.940, 1.069) | 0.895 |
| No. of preoperative medications | 2.928 (0.810, 12.47) | 0.134 |
| Preoperative visual acuity, logMAR | 0.747 (0.091, 2.745) | 0.721 |
| Previous cataract extraction | 5.769 (1.365, 39.81) | 0.032 |
| Central corneal thickness, μm | 1.004 (0.99, 1.018) | 0.591 |
| Axial length, mm | 1.351 (1.005, 1.883) | 0.052 |
| Visual field index, % | 0.996 (0.98, 1.016) | 0.684 |
| Mean deviation, dB | 0.98 (0.915, 1.053) | 0.621 |
| Pattern standard deviation, dB | 1.1 (0.928, 1.326) | 0.288 |

CI, confidence interval; IOP, intraocular pressure; logMAR, logarithm of the minimum angle of resolution.

duration between surgeries ranged from 6 to 321 months and the average was 76 ± 73 months. The duration was not associated with surgical failure in univariate logistic regression analysis (odds ratio = 0.999, $P$ = 0.800). All intrableb parameters were not related to the duration ($P \geq 0.297$ for all) except for microcyst formation ($r$ = 0.353, $P$ = 0.035).

## Representative cases after trabeculectomy with AMT

Fig 5A–5C shows the slit-lamp photographs and intrableb structures of a successful case of trabeculectomy with AMT. Slit-lamp examination showed a medium height, mildly vascular bleb

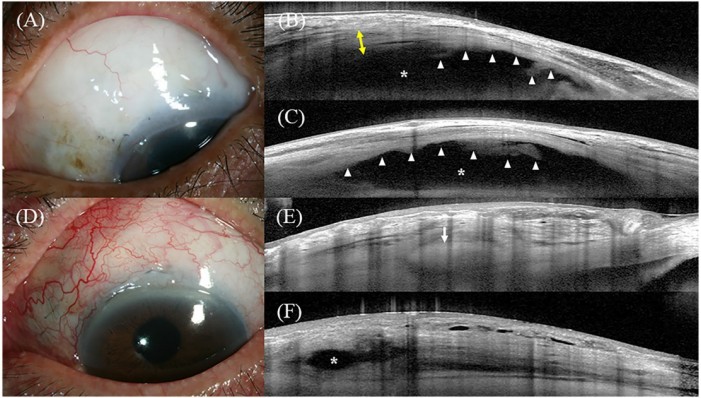

**Fig 5. Representative images of AS-OCT.** Fig A-C represent a successful case and Fig D-F represent a failed case. (A) Slit-lamp photo of the bleb at 19 months after trabeculectomy with AMT. The IOP preoperatively was 21 mmHg and at the time of the AS-OCT examination was 7 mmHg. (B) Vertical AS-OCT scan of the bleb. The AS-OCT showed a tall bleb with a hyporeflective striping layer (*yellow two-way arrows*). Fluid-filled space (*asterisk*) extended posteriorly beyond the view of the image field. Amniotic membrane was noted even >19 months after surgery (*white arrow head*). (C) Horizontal AS-OCT scan of the bleb. The AS-OCT showed a diffuse fluid-filled space with the amniotic membrane (*white arrow head*). (D) Slit-lamp photo of the bleb at 6 months after trabeculectomy with AMT. The IOP preoperatively was 23 mmHg and at the time of the AS-OCT examination was 19 mmHg. (E) Vertical and (F) Horizontal AS-OCT scans of the bleb. The *white arrow* indicates the scleral flap. The bleb height was low, with limited fluid-filled space (*asterisk*) and no striping layer. AS-OCT, anterior segment optical coherence tomography; AMT, amniotic membrane transplantation; IOP, intraocular pressure.

at the 3 o'clock hours width, based on the Indiana Bleb Appearance Grading Scale (H2 E2 V2 S0) (Fig 5A) [27]. AS-OCT performed 19 months after the surgery showed that the fluid-filled space extended beyond the view of the image field and the amniotic membrane was noted beneath the bleb wall in both vertical and horizontal scans (FFSS 2) (Fig 5B and 5C).

Fig 5D–5F shows slit-lamp photographs and intrableb structures of a failed case of trabeculectomy with AMT. Slit-lamp examination revealed a flat bleb with moderate vascularity (H0 E0 V3 S0) (Fig 5D). AS-OCT at six months after surgery showed that there was limited fluid-filled space in the vertical (FFSS 0) and horizontal (FFSS 1) scans with no striping layer (Fig 5E and 5F).

## Discussion

To the best of our knowledge, this is the first study to evaluate the AS-OCT findings of intrableb structures after trabeculectomy with AMT. In this study, the quantitative AS-OCT parameters of the intrableb structures of the success group were significantly different from those of the failure group. We found that the success group had a taller bleb with greater FFSS, a thicker bleb wall, and a striping layer than the failure group did. Microcyst formation was more frequently seen in the success group than in the failure group, whereas the bleb wall reflectivity in the success group was lower than that in the failure group. The results were consistent when the effect of the AS-OCT examination time was adjusted.

Our findings are consistent with the results of previous studies on the association between intrableb structures and bleb function. Narita et al. [21] evaluated intrableb structures one year after trabeculectomy using AS-OCT. They reported that taller blebs were associated with good IOP control. In their follow-up study, the striping phenomenon at 2 weeks post-trabeculectomy predicted good IOP control at 1 year postoperatively [20]. Theelen et al. [26] found that a hyporeflective striping layer corresponded to a drainage channel, suggesting the importance of this layer in a successful bleb. Hamanaka et al. [23] investigated the relationship between bleb morphology and IOP control using AS-OCT after limbal- and fornix-based trabeculectomy. The combination of bleb height, extent, and minimum bleb wall thickness were significantly correlated with IOP control in the limbal-based trabeculectomy group, while ciliochoroidal detachment and lake under the scleral flap were significantly correlated with IOP control in the fornix-based trabeculectomy group [23].

The results of this study were consistent with our hypothesis and those of previous studies [20, 21, 23, 26], except for the fluid-filled space height. We speculated that this may be due to differences in the outcome measures. Previous studies did not divide the bleb height into bleb wall, striping layer, and fluid-filled space height [20, 21]. In addition, encysted blebs were observed in three of the 12 patients in the failure group in our study. The encysted bleb might have influenced the bleb height to be measured tall, even though it was considered a failing bleb [28].

Only one study has reported on intrableb structures after trabeculectomy with AMT [18]. Nakamura et al. [18] compared the intrableb structure after trabeculectomy between AMT and non-AMT groups using UBM. They reported that a posteriorly extending fluid-filled space rather than reflectivity in the AMT group was associated with long-term IOP control. Our findings are consistent with those of their study [18]. In our study, the fluid-filled space of the success group extended more posteriorly than that of the failure group did. In addition, quantitative assessment of the bleb wall reflectivity revealed that it was significantly lower in the success group than in the failure group. After trabeculectomy, the aqueous humor flows out from the edge of the scleral flap and enters the subconjunctival space. The aqueous humor is absorbed by conjunctival blood vessels or lymphatics or crosses the conjunctiva into the tear

layer [29, 30]. Based on the AS-OCT findings of this study, the transconjunctival and subconjunctival pathways may play a role in aqueous humor outflow after trabeculectomy with AMT. In this study, we found that the striping layer, which are fluid-filled channels with fine connective tissue inside the Tenon's layer that constitute the transconjunctival filtration, was important for maintaining well-functioning blebs, as reported previously [20, 24, 26]. More frequent microcyst formation in the success group also supports the importance of transconjunctival filtration after trabeculectomy with AMT. Furthermore, the amniotic membrane is semi-permeable to water and can reduce water loss by maintaining a physiologically moist environment [31, 32].

Although there was no difference between the fluid-filled space heights of the success and failure groups, the significant difference in FFSS indicated that absorption through the subconjunctival pathway plays a role in aqueous humor outflow. The amniotic membrane may prevent adhesion between the conjunctiva and sclera and subsequently help aqueous humor flow posteriorly. The amniotic membrane exhibits anti-fibrotic properties by inhibiting the proliferation of conjunctival fibroblasts and differentiation of fibroblasts into myofibroblasts [33]. Amniotic membrane regulates fibrotic reactions by suppressing TGF-β signaling, whose levels are elevated in patients with POAG [34–36]. Furthermore, scarring following glaucoma filtration surgery activates inflammatory cytokines including TGF-β [37].

In the univariate logistic regression analysis, previous cataract surgery had a significant impact on surgical failure among various factors, such as age, sex, preoperative IOP, preoperative visual acuity, CCT, axial length, and visual field parameters. This may be explained by the fact that surgical incision-induced conjunctival scarring and alterations in the aqueous humor nature may contribute to the failure of trabeculectomy [38, 39]. In the subgroup analysis for pseudophakic eyes, the duration between cataract extraction and trabeculectomy with AMT was not significantly associated with surgical failure. In the correlation analysis, the duration was positively correlated to microcyst formation. Damage of blood aqueous barrier after cataract extraction surgery is likely to persist for a long period of time [40]. However, it is also possible to recover over time, suggesting that a positive correlation between the duration and microcyst formation [41].

We used a stringent criterion for success of trabeculectomy with AMT in this study. Previous studies suggested that IOP < 21 mmHg may not be optimal success criteria for trabeculectomy [42, 43]. Recent clinical trials have adopted cut off points of IOP < 18 mmHg based on long-term outcomes of glaucoma surgery [44, 45].

Narita et al. [20] compared AS-OCT parameters between 2 weeks and 1 year after trabeculectomy and found that bleb height, bleb wall thickness and ratio of hyporeflective space of the bleb wall tended to decrease in the unsuccessful IOP control group while bleb height and ratio of hyporeflective space of the bleb wall significantly increased in the successful IOP control group at 1 year post-trabeculectomy. With regard to qualitative AS-OCT parameter, the appearance ratios of cyst-like structures and striping phenomenon decreased in both groups. Dangda et al. [46] also reported that reduction in microcysts were noted at intermediate (6 to 12 months) and long-term (>12 months) timepoints compared to early postoperative period (≤ 3 months) after sub-Tenon's implantation of the Xen Gel stent. As longitudinal changes in the intrableb structure after trabeculectomy with AMT was not assessed in this study, further research is needed in order to evaluate longitudinal changes in the intrableb structures with AS-OCT after trabeculectomy with AMT.

Our study had some limitations. First, the interval between surgery and AS-OCT examination varied from patient to patient. As this was a retrospective study, there was no coincidence in the timing of AS-OCT examination among patients. To evaluate the difference over time, a linear regression model was used to test the interaction effect, and there was no significant

interaction effect between the time and groups. We found that the results were consistent when the AS-OCT examination time was adjusted. In addition, although the examination time was different for each patient, AS-OCT images were taken at least six months after trabeculectomy. Second, a relatively small number of patients were included, as this was a retrospective study conducted at a single center. Third, longitudinal changes in the intrableb structure after trabeculectomy with AMT was not assessed in this study. Fourth, we did not compare the intrableb structures between trabeculectomy with AMT group and without AMT group. Future studies that overcome these limitations are required.

Using AS-OCT, we demonstrated significant differences in the structure between functioning and non-functioning blebs after trabeculectomy with AMT. We found that a posteriorly extending fluid-filled space, a tall bleb with low bleb wall reflectivity, and a thick striping layer may be features of a successful filtering bleb after trabeculectomy with AMT.

## Supporting information

**S1 Fig. Scatter plot with a simple linear regression line.** There were no significant interaction effects between the AS-OCT examination time and groups in all the bleb parameters ($P \geq 0.333$ for all).
(TIF)

## Acknowledgments

We would like to thank Editage (www.editage.co.kr) for the English language editing.

## Author Contributions

**Conceptualization:** Jiwoong Lee.

**Data curation:** Jinmi Kim, Jiwoong Lee.

**Formal analysis:** Hwayeong Kim, Sangwoo Moon, Eunah Kim, Jiwoong Lee.

**Funding acquisition:** Jiwoong Lee.

**Investigation:** Hwayeong Kim, Jiwoong Lee.

**Methodology:** Jinmi Kim, Jiwoong Lee.

**Project administration:** Jiwoong Lee.

**Resources:** Jiwoong Lee.

**Supervision:** Jiwoong Lee.

**Validation:** Jiwoong Lee.

**Visualization:** Hwayeong Kim, Jiwoong Lee.

**Writing – original draft:** Hwayeong Kim, Jiwoong Lee.

**Writing – review & editing:** Hwayeong Kim, Sangwoo Moon, Eunah Kim, Jiwoong Lee.

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
