## [Decision Letter · Decision Letter 0]

26 Jan 2023

PONE-D-22-33214Bleb analysis using anterior segment optical coherence tomography after trabeculectomy with amniotic membrane transplantationPLOS ONE

Dear Dr. Lee,

Thank you for submitting your manuscript to PLOS ONE. After careful consideration, we feel that it has merit but does not fully meet PLOS ONE’s publication criteria as it currently stands. Therefore, we invite you to submit a revised version of the manuscript that addresses the points raised during the review process.

The reviewers, in general, find your MSS appropriate but expressed few reasonable concerns. Thus, MSS requires adequate addressing of points given below prior to publication decision. 

We look forward to receiving your revised manuscript.

Kind regards,

Rajiv R. Mohan, Ph.D.

Academic Editor

PLOS ONE

Journal Requirements:

"This research was supported by the Patient Centered Clinical Research Coordinating Center, funded by the Ministry of Health & Welfare, Republic of Korea (grant no. HI19C0481, HC19C0276)."

3. Please include a caption for figure 5. 

Additional Editor Comments:

Dear authors,

The reviewers, in general, find your MSS appropriate but expressed few reasonable concerns. Thus, MSS requires adequate addressing of points given below prior to publication decision.

Reviewers' comments:

Reviewer's Responses to Questions

**Comments to the Author**

1. Is the manuscript technically sound, and do the data support the conclusions?

Reviewer #1: Yes

Reviewer #2: Yes

2. Has the statistical analysis been performed appropriately and rigorously? 

Reviewer #1: Yes

Reviewer #2: Yes

3. Have the authors made all data underlying the findings in their manuscript fully available?

Reviewer #1: Yes

Reviewer #2: Yes

4. Is the manuscript presented in an intelligible fashion and written in standard English?

Reviewer #1: Yes

Reviewer #2: Yes

5. Review Comments to the Author

Reviewer #1: The manuscript ID: PONE-D-22-33214; entitled “Bleb analysis using anterior segment optical coherence tomography after trabeculectomy with amniotic membrane transplantation” is an interesting and great peer article in the ocular research field The article highlights the potential of AS-OCT imaging techniques as a tool for bleb analysis after surgery. The article covers the basic and translational fundamentals of imaging techniques with enough patient data. The author must be revised the comments for a better understanding of the readers in some of the sections as details mentioned below:

Comment 1: In the abstract section, add one line about “why you are doing this work and the importance of the work” and then come to the objective of the study.

Comment 2: In the introduction section, page number 3, lines 44-45 add the other fibrosis regulators like epigenetic modification (SAHA), etc. contribution, and limitation. See the reference below:

Ref. “Sharma et al 2016. Epigenetic Modification Prevents Excessive Wound Healing and Scar Formation After Glaucoma Filtration Surgery. Invest Ophthalmol Vis Sci. 2016 Jun 1;57(7):3381-9. doi: 10.1167/iovs.15-18750. PMID: 27367506; PMCID: PMC4961058”.

Comment 3: In the introduction section, page number 4, line 77 has to restructured due to this is not the first study. See the reference below:

Ref. “Kim H, Moon S, Kim J, Lee J. The Effect of Amniotic Membrane Transplantation on Trabeculectomy in Patients with Pseudoexfoliation Glaucoma. J Ophthalmol. 2022 Jul 30;2022:9355206. doi: 10.1155/2022/9355206. PMID: 35942064; PMCID: PMC9356778.”.

Comment 4: In the result section, page number 14, lines 248-250, define the figure 4 legend each box plots to A-G, for better clarity for the readers.

Comment 5: In the result section, page number 15, lines 265-290, the figure number not matched the figure (Figure 4 should be Figure 5). I think, the authors are explaining figure 5, and due to typos or accidental errors, they have mentioned figure 4 in every place rather than figure 5.

Comment 6: In the discussion section, page number 16, line 293 has to restructured due to this is not the first study. See the reference below:

Ref. “Kim H, Moon S, Kim J, Lee J. The Effect of Amniotic Membrane Transplantation on Trabeculectomy in Patients with Pseudoexfoliation Glaucoma. J Ophthalmol. 2022 Jul 30;2022:9355206. doi: 10.1155/2022/9355206. PMID: 35942064; PMCID: PMC9356778.”.

In summary, the review article ID: ID: PONE-D-22-33214; entitled “Bleb analysis using anterior segment optical coherence tomography after trabeculectomy with amniotic membrane transplantation” is good and worth publication in the PlosOne journal after addressing these minor comments.

Reviewer #2: This research was about anterior segment optical coherence tomographic findings after trabeculectomy with amniotic membrane transplantation to find out bleb characteristics by using mainly anterior segment OCT and slit-lamp and authors finally searched surgical success predictor.

The topic is very interesting and useful for readers who are Glaucoma surgeon and read PLOSONE. Moreover as authors described in this manuscript already, wound healing and wound modulation is quite important in management of glaucoma.

However, there are a few studies regarding AS-OCT after trabeculectomy with AMT. Therefore, this manuscript look meaningful study because it provides clinically significant important information with AS-OCT.

However, the study has some mild concerns and requires minor revisions and clarification in order to meet the journal's requirements.

1. As described by the authors, the timing of anterior segment OCT is quite different. Please provide the range of the timing of anterior segment OCT in Table 1 (If possible, please provide the routine exam timing in usual clinical settings.)

2. Please provide the reason that surgical success was defined as IOP < 18 mmHg?

3. And if possible, please describe or discuss the clinical postoperative changes of anterior segment OCT. (In usual, the bleb height might be lower or other morphological changes can be occurred during long-term follow-up)

Anyway, this manuscript has clinical relevant data, and looks technically sound. Thank you for your great contribution to the field of surgical glaucoma.

6. PLOS authors have the option to publish the peer review history of their article (what does this mean?). If published, this will include your full peer review and any attached files.

Reviewer #1: **Yes: **Suneel Gupta

Reviewer #2: No

---

## [Author Response · Author response to Decision Letter 0]

5 Feb 2023

Dear, Prof. Rajiv R. Mohan.

Academic Editor

PLOS ONE

Attached please find the revised manuscript entitled “Bleb analysis using anterior segment optical coherence tomography after trabeculectomy with amniotic membrane transplantation”. A point-by-point response to each of reviewer’s comments follows. The authors wish to thank the Editorial Board for their thoughtful consideration and recommendations, and hope that this revised manuscript now meets your requirements for publication.

Reviewer #1: The manuscript ID: PONE-D-22-33214; entitled “Bleb analysis using anterior segment optical coherence tomography after trabeculectomy with amniotic membrane transplantation” is an interesting and great peer article in the ocular research field. The article highlights the potential of AS-OCT imaging techniques as a tool for bleb analysis after surgery. The article covers the basic and translational fundamentals of imaging techniques with enough patient data. The author must be revised the comments for a better understanding of the readers in some of the sections as details mentioned below:

Comment 1: In the abstract section, add one line about “why you are doing this work and the importance of the work” and then come to the objective of the study.

Response: We added the following background of this study in the abstract section.

“Little has been known about the intrableb structures associated with bleb function after trabeculectomy with amniotic membrane transplantation (AMT).”

Comment 2: In the introduction section, page number 3, lines 44-45 add the other fibrosis regulators like epigenetic modification (SAHA), etc. contribution, and limitation. See the reference below:

Ref. “Sharma et al 2016. Epigenetic Modification Prevents Excessive Wound Healing and Scar Formation After Glaucoma Filtration Surgery. Invest Ophthalmol Vis Sci. 2016 Jun 1;57(7):3381-9. doi: 10.1167/iovs.15-18750. PMID: 27367506; PMCID: PMC4961058”.

Response: We added the other fibrosis regulators in the introduction section as belows:

“An experimental study reported that suberoylanilide hydroxamic acid, a histone deacetylase inhibitor prevented excessive wound healing and scar formation in a rabbit model of glaucoma filtration surgery.[9]” (Invest Ophthalmol Vis Sci. 2016;57:3381–3389.)

Comment 3: In the introduction section, page number 4, line 77 has to restructured due to this is not the first study. See the reference below:

Ref. “Kim H, Moon S, Kim J, Lee J. The Effect of Amniotic Membrane Transplantation on Trabeculectomy in Patients with Pseudoexfoliation Glaucoma. J Ophthalmol. 2022 Jul 30;2022:9355206. doi: 10.1155/2022/9355206. PMID: 35942064; PMCID: PMC9356778.”.

Response: The bleb analysis with anterior segment optical coherence tomography was not assessed in the previous study of our group (J Ophthalmol. 2022 Jul 30;2022:9355206.). We mentioned the previous study in the introduction section along with another study from our study group. 

Comment 4: In the result section, page number 14, lines 248-250, define the figure 4 legend each box plots to A-G, for better clarity for the readers.

Response: We added the following explanation for each Figure 4A to G. 

“(A) Bleb was higher in the success group than that in the failure group. (B) Bleb wall was thicker in the success group than that in the failure group. (C) Stripping layer was thicker in the success groups than that in the failure group. (D) Bleb wall reflectivity was lower in the success group than that in the failure group. (E) Fluid-filled space score was greater than that in the failure group. (F) There was no significant difference in the fluid-filled space height between the two groups. (G) Microcyst formation was more frequently noted in the success group than in the failure group.”

Comment 5: In the result section, page number 15, lines 265-290, the figure number not matched the figure (Figure 4 should be Figure 5). I think, the authors are explaining figure 5, and due to typos or accidental errors, they have mentioned figure 4 in every place rather than figure 5.

Response: We corrected the figure number from 4 to 5.

Comment 6: In the discussion section, page number 16, line 293 has to restructured due to this is not the first study. See the reference below:

Ref. “Kim H, Moon S, Kim J, Lee J. The Effect of Amniotic Membrane Transplantation on Trabeculectomy in Patients with Pseudoexfoliation Glaucoma. J Ophthalmol. 2022 Jul 30;2022:9355206. doi: 10.1155/2022/9355206. PMID: 35942064; PMCID: PMC9356778.”.

Response: The bleb analysis with anterior segment optical coherence tomography was not assessed in the previous study of our group (J Ophthalmol. 2022 Jul 30;2022:9355206.). We mentioned the previous study in the introduction section along with another study from our study group. 

Reviewer #2: This research was about anterior segment optical coherence tomographic findings after trabeculectomy with amniotic membrane transplantation to find out bleb characteristics by using mainly anterior segment OCT and slit-lamp and authors finally searched surgical success predictor.

The topic is very interesting and useful for readers who are Glaucoma surgeon and read PLOSONE. Moreover as authors described in this manuscript already, wound healing and wound modulation is quite important in management of glaucoma.

However, there are a few studies regarding AS-OCT after trabeculectomy with AMT. Therefore, this manuscript look meaningful study because it provides clinically significant important information with AS-OCT.

However, the study has some mild concerns and requires minor revisions and clarification in order to meet the journal's requirements.

Comment 1. As described by the authors, the timing of anterior segment OCT is quite different. Please provide the range of the timing of anterior segment OCT in Table 1 (If possible, please provide the routine exam timing in usual clinical settings.)

Response: The interval between surgery and AS-OCT examination ranged from 6 to 39 months (median, 19 months). We added the range of anterior segment OCT examination time in Table 1.

Comment 2. Please provide the reason that surgical success was defined as IOP < 18 mmHg?

Response: We used a stringent criterion for success of trabeculectomy with amniotic membrane transplantation in this study. Previous studies suggested that IOP < 21 mmHg may not be optimal success criteria for trabeculectomy (Ophthalmology 2006;113:930 –936) (J Glaucoma 2017;26:303–310). Recent clinical trials have adopted cut off points of IOP < 18 mmHg based on long-term outcomes of glaucoma surgery (Am J Ophthalmol. 2000 Oct;130(4):429-40.) (Ophthalmology. 2001 Nov;108(11):1943-53.).

We added the above reason why surgical success was defined as IOP < 18 mmHg in the discussion section. 

“We used a stringent criterion for success of trabeculectomy with AMT in this study. Previous studies suggested that IOP < 21 mmHg may not be optimal success criteria for trabeculectomy.[40,41] Recent clinical trials have adopted cut off points of IOP < 18 mmHg based on long-term outcomes of glaucoma surgery.[42,43]”

Comment 3. And if possible, please describe or discuss the clinical postoperative changes of anterior segment OCT. (In usual, the bleb height might be lower or other morphological changes can be occurred during long-term follow-up)

Response: There are a few studies which analyzed longitudinal changes in intrableb structure using AS-OCT after glaucoma surgery (Br J Ophthalmol 2018;102:796–801.) (J Glaucoma. 2021;30:988-995.). Narita et al. compared AS-OCT parameters between 2 weeks and 1 year after trabeculectomy and found that bleb height, bleb wall thickness and ratio of hyporeflective space of the bleb wall tended to decrease in the unsuccessful IOP control group while bleb height and ratio of hyporeflective space of the bleb wall significantly increased in the successful IOP control group at 1 year post-trabeculectomy (Br J Ophthalmol 2018;102:796–801.). With regard to qualitative AS-OCT parameter, the appearance ratios of cyst-like structures and striping phenomenon decreased in both groups (Br J Ophthalmol 2018;102:796–801.). Dangda et al. also reported that reduction in microcysts were noted at intermediate (6 to 12 months) and long-term (>12 months) timepoints compared to early postoperative period (≤ 3 months) after Sub-Tenon's implantation of the Xen Gel stent (J Glaucoma. 2021;30:988-995.). In this study, however, longitudinal changes in the intrableb structure after trabeculectomy with AMT was not assessed. Therefore, further research is needed in order to evaluate longitudinal changes in the intrableb structures with AS-OCT after trabeculectomy with AMT.”

We added following sentences in the limitation section.

“Third, longitudinal changes in the intrableb structure after trabeculectomy with AMT was not assessed in this study. Narita et al.[20] compared AS-OCT parameters between 2 weeks and 1 year after trabeculectomy and found that bleb height, bleb wall thickness and ratio of hyporeflective space of the bleb wall tended to decrease in the unsuccessful IOP control group while bleb height and ratio of hyporeflective space of the bleb wall significantly increased in the successful IOP control group at 1 year post-trabeculectomy. With regard to qualitative AS-OCT parameter, the appearance ratios of cyst-like structures and striping phenomenon decreased in both groups. Dangda et al.[44] also reported that reduction in microcysts were noted at intermediate (6 to 12 months) and long-term (>12 months) timepoints compared to early postoperative period (≤ 3 months) after sub-Tenon's implantation of the Xen Gel stent. Further research is needed in order to evaluate longitudinal changes in the intrableb structures with AS-OCT after trabeculectomy with AMT.”

Sincerely yours,

Jiwoong Lee, MD, PhD. 

Department of Ophthalmology, Pusan National University College of Medicine

#179 Gudeok-ro, Seo-gu, Busan 49241, Korea

Tel: +82-51-240-7958

Fax: +82-51-242-7341

Email: alertlee@naver.com

---

## [Decision Letter · Decision Letter 1]

30 Mar 2023

PONE-D-22-33214R1Bleb analysis using anterior segment optical coherence tomography after trabeculectomy with amniotic membrane transplantationPLOS ONE

Dear Dr. Lee,

Thank you for submitting your manuscript to PLOS ONE. After careful consideration, we feel that it has merit but does not fully meet PLOS ONE’s publication criteria as it currently stands. Therefore, we invite you to submit a revised version of the manuscript that addresses the points raised during the review process.

 The authors present a well written manuscript describing the bleb structure by AS-OCT after Trab with amniotic membrane. The manuscript in its current format is well written and the authors are to congratulated for excellent quality photos. A few comments remaining include: Please add to the results section, the duration between cataract surgery and the Trab surgery in the eyes that were pseudophakic, if this information was available and study the effect of this duration on the success-failure of the procedure and bleb appearance. Please move the part of the literature review (in the limitations of the study) to the body of the discussion section, before the limitations of the study, and restrict the limitations to just mentioning the fact that "Third, longitudinal changes in the intrableb structure after trabeculectomy with AMT was not assessed in this study."  Please add to the limitations of the study that it was not compared to eyes subjected to Trab without AMT, if it was the target of the authors to demonstrate the additive effect of AMT on Trab with MMC for POAG.

We look forward to receiving your revised manuscript.

Kind regards,

Nader Hussien Lotfy Bayoumi, M.D., FRCS (Glasgow)

Academic Editor

PLOS ONE

Journal Requirements:

Reviewers' comments:

Reviewer's Responses to Questions

**Comments to the Author**

1. If the authors have adequately addressed your comments raised in a previous round of review and you feel that this manuscript is now acceptable for publication, you may indicate that here to bypass the “Comments to the Author” section, enter your conflict of interest statement in the “Confidential to Editor” section, and submit your "Accept" recommendation.

Reviewer #1: All comments have been addressed

Reviewer #2: All comments have been addressed

2. Is the manuscript technically sound, and do the data support the conclusions?

Reviewer #1: Yes

Reviewer #2: Yes

3. Has the statistical analysis been performed appropriately and rigorously? 

Reviewer #1: Yes

Reviewer #2: Yes

4. Have the authors made all data underlying the findings in their manuscript fully available?

Reviewer #1: Yes

Reviewer #2: Yes

5. Is the manuscript presented in an intelligible fashion and written in standard English?

Reviewer #1: Yes

Reviewer #2: Yes

6. Review Comments to the Author

Reviewer #1: (No Response)

Reviewer #2: Thank you for point-by-point revisions.

The authors well descrbied the issues raised by reviewers.

Thank you again for wonderful contributions.

7. PLOS authors have the option to publish the peer review history of their article (what does this mean?). If published, this will include your full peer review and any attached files.

Reviewer #1: **Yes: **Suneel Gupta

Reviewer #2: No

---

## [Author Response · Author response to Decision Letter 1]

13 Apr 2023

Dear, Nader Hussien Lotfy Bayoumi

Academic Editor

PLOS ONE

Attached please find the revised manuscript entitled “Bleb analysis using anterior segment optical coherence tomography after trabeculectomy with amniotic membrane transplantation”. A point-by-point response to the comments follows. The authors wish to thank the Editorial Board for their thoughtful consideration and recommendations, and hope that this revised manuscript now meets your requirements for publication.

Comment 1: Please add to the results section, the duration between cataract surgery and the Trab surgery in the eyes that were pseudophakic, if this information was available and study the effect of this duration on the success-failure of the procedure and bleb appearance.

Response: We investigated the duration between cataract extraction surgery and the trabeculectomy with AMT and evaluated the effect of this duration on the surgical outcome and intrableb parameters. We added the following sentences in the Result section.

“The effect of duration between cataract extraction and trabeculectomy with AMT on surgical failure and relationship between the duration and intrableb structures were evaluated. The duration between surgeries ranged from 6 to 321 months and the average was 76 ± 73 months. The duration was not associated with surgical failure in univariate logistic regression analysis (odds ratio = 0.999, P = 0.800). All intrableb parameters were not related to the duration (P ≥ 0.297 for all) except for microcyst formation (r = 0.353, P = 0.035).”

We also added following sentences in the Discussion section.

“In the subgroup analysis for pseudophakic eyes, the duration between cataract extraction and trabeculectomy with AMT was not significantly associated with surgical failure. In the correlation analysis, the duration was positively correlated to microcyst formation. Damage of blood aqueous barrier after cataract extraction surgery is likely to persist for a long period of time.[40] (Eye, 2000, 14.1: 61-63.) However, it is also possible to recover over time, suggesting that a positive correlation between the duration and microcyst formation.[41] (British Journal of Ophthalmology. 1991;75: 106–110.)"

Comment 2: Please move the part of the literature review (in the limitations of the study) to the body of the discussion section, before the limitations of the study, and restrict the limitations to just mentioning the fact that "Third, longitudinal changes in the intrableb structure after trabeculectomy with AMT was not assessed in this study." 

Response: We moved the part of the literature review before the limitation section shown as below.

“Narita et al.[20] compared AS-OCT parameters between 2 weeks and 1 year after trabeculectomy and found that bleb height, bleb wall thickness and ratio of hyporeflective space of the bleb wall tended to decrease in the unsuccessful IOP control group while bleb height and ratio of hyporeflective space of the bleb wall significantly increased in the successful IOP control group at 1 year post-trabeculectomy. With regard to qualitative AS-OCT parameter, the appearance ratios of cyst-like structures and striping phenomenon decreased in both groups. Dangda et al.[46] also reported that reduction in microcysts were noted at intermediate (6 to 12 months) and long-term (>12 months) timepoints compared to early postoperative period (≤ 3 months) after sub-Tenon's implantation of the Xen Gel stent. As longitudinal changes in the intrableb structure after trabeculectomy with AMT was not assessed in this study, further research is needed in order to evaluate longitudinal changes in the intrableb structures with AS-OCT after trabeculectomy with AMT.”

We restricted the third limitation shown as below.

“Third, longitudinal changes in the intrableb structure after trabeculectomy with AMT was not assessed in this study.”

Comment 3: Please add to the limitations of the study that it was not compared to eyes subjected to Trab without AMT, if it was the target of the authors to demonstrate the additive effect of AMT on Trab with MMC for POAG.

Response: Although the purpose of this study was not to demonstrate the additive effect of AMT on trabeculectomy with MMC for patients with POAG, we added following sentences in limitation section.

“Fourth, we did not compare the intrableb structures between trabeculectomy with AMT group and without AMT group.”

Sincerely yours,

Jiwoong Lee, MD, PhD. 

Department of Ophthalmology, Pusan National University College of Medicine

#179 Gudeok-ro, Seo-gu, Busan 49241, Korea

Tel: +82-51-240-7958

Fax: +82-51-242-7341

Email: alertlee@naver.com

---

## [Editor Report · Decision Letter 2]

16 Apr 2023

Bleb analysis using anterior segment optical coherence tomography after trabeculectomy with amniotic membrane transplantation

PONE-D-22-33214R2

Dear Dr. Lee,

We’re pleased to inform you that your manuscript has been judged scientifically suitable for publication and will be formally accepted for publication once it meets all outstanding technical requirements.

Kind regards,

Nader Hussien Lotfy Bayoumi, M.D., FRCS (Glasgow)

Academic Editor

PLOS ONE

Additional Editor Comments:

Thank you for the response to the reviewers' comments.
---

## [Editor Report · Acceptance letter]

24 Apr 2023

PONE-D-22-33214R2 

Bleb analysis using anterior segment optical coherence tomography after trabeculectomy with amniotic membrane transplantation 

Dear Dr. Lee:

I'm pleased to inform you that your manuscript has been deemed suitable for publication in PLOS ONE. Congratulations! Your manuscript is now with our production department. 

Kind regards, 

on behalf of

Professor Nader Hussien Lotfy Bayoumi 

Academic Editor

PLOS ONE